SOFTWARE

# RNAtango: Analysing and comparing RNA 3D structures via torsional angles

**Marta Mackowiak[1], Bartosz Adamczyk[1], Marta Szachniuk**  [1,2]*, **Tomasz Zok[1]***

**1** Institute of Computing Science, Poznan University of Technology, Poznan, Poland, **2** Institute of Bioorganic Chemistry, Polish Academy of Sciences, Poznan, Poland

* mszachniuk@cs.put.poznan.pl (MS); tzok@cs.put.poznan.pl (TZ)

## Abstract

RNA molecules, essential for viruses and living organisms, derive their pivotal functions from intricate 3D structures. To understand these structures, one can analyze torsion and pseudo-torsion angles, which describe rotations around bonds, whether real or virtual, thus capturing the RNA conformational flexibility. Such an analysis has been made possible by RNAtango, a web server introduced in this paper, that provides a trigonometric perspective on RNA 3D structures, giving insights into the variability of examined models and their alignment with reference targets. RNAtango offers comprehensive tools for calculating torsion and pseudo-torsion angles, generating angle statistics, comparing RNA structures based on backbone torsions, and assessing local and global structural similarities using trigonometric functions and angle measures. The system operates in three scenarios: single model analysis, model-versus-target comparison, and model-versus-model comparison, with results output in text and graphical formats. Compatible with all modern web browsers, RNAtango is accessible freely along with the source code. It supports researchers in accurately assessing structural similarities, which contributes to the precision and efficiency of RNA modeling.

**Data Availability Statement:** • RNAtango (RNA structure analysis in Torsion ANGle dOmain) is a web-based tool accessible at https://rnatango.cs.put.poznan.pl It is free and available to all users without logging in. The rich and ergonomic user

## Introduction

RNA molecules play a crucial role in all living organisms by supervising critical cellular processes [1]. Understanding these capabilities requires studying their complex 3D structures using a range of experimental and computational techniques [2–6]. The latter focus on various properties of the structures, as made apparent by the corresponding representations of structural information applied in the output. An effective method for decoding the 3D architecture of RNA is the trigonometric representation, which uses torsion and pseudo-torsion angles to draw a detailed picture of the molecule [7]. What are those angles? If we look at RNA as a kind of flexible chain, we can see that each link in the chain can rotate around incident bonds. These rotations are measured by torsion angles, which describe the flexibility of the backbone, the conformation of ribose, and the alignment of the glycosidic bond that connects sugar to the nucleobase [8, 9]. Furthermore, pseudo-torsion angles complete the view of the structural landscape of RNA, enriching our understanding of its 3D conformation [10–13].

interface of the system supports all modern web browsers, platforms, and mobile devices, and provides a variety of downloadable data in text and graphical formats. RNAtango incorporates two algorithms to compute the angular similarity of RNA structures, MCQ and LCS-TA. They are available as a separate Java application named MCQ4Structures, which can be downloaded from https://github.com/tzok/mcq4structures. The source codes of RNAtango are available at https://github.com/tzok/RNAtango. All other relevant data are within the manuscript and its Supporting information file.

**Funding:** We acknowledge funding from the National Science Center, Poland (https://www.ncn.gov.pl/en) [grant 2023/51/D/ST6/01207 to TZ], Poznan University of Technology (https://www.put.poznan.pl/en) and the Institute of Bioorganic Chemistry, PAS (https://www.ibch.poznan.pl/en.html) [statutory funds]. The funders had no role in study design, data collection and analysis, decision to publish, or preparation of the manuscript.

**Competing interests:** The authors have declared that no competing interests exist.

Unlike Cartesian coordinates, which describe the spatial positions of atoms, angular parameters represent the rotational relationships between bonds. Specific torsional angles can be associated with functional RNA motifs, such as binding sites to proteins, small molecules, or other RNAs [15–18]. As a result, angular analysis aids in predicting and verifying functional sites, which is crucial for understanding the role of RNA in many biological processes. Additionally, RNA folding is governed by the molecule's complex energetic landscape, shaped by factors such as bonded and non-bonded interactions, as well as planar and torsional angles. These angular relationships play a pivotal role in RNA structure, leading to a growing adoption of trigonometric representations in computational methods aimed at predicting the 3D structure of RNA [16, 19–26]. From a computational perspective, angular parameters simplify the comparison of different RNA conformations by reducing the complexity of structural data into a series of rotational values, making it easier to identify similarities and differences between RNA structures. However, the lower granularity of trigonometric representations can make it more challenging to analyze structures at the single-atom level. Therefore, to gain a more comprehensive understanding of structural similarities and differences, it is recommended to use both atomic coordinate-based and torsion angle-based methods in comparative analyses.

Several bioinformatics tools have been designed for analyzing torsion and pseudotorsion angles, each with its own strengths and limitations. 3DNA, an open-source toolkit, provides comprehensive functionality, including torsion and pseudotorsion angle calculations [27], but lacks support for the current standard PDBx/mmCIF file format. DSSR, the successor to 3DNA, overcomes this limitation by supporting both PDB and PDBx/mmCIF files. However, it is a closed-source, commercial application that requires licensing, even for research purposes [28]. Curves+, another tool used for torsion angle analysis, is currently inaccessible due to the unavailability of its webpage and source code hosting [29]. Barnaba, a Python library and toolset for analyzing single structures or trajectories, supports torsion angle calculations but, like 3DNA, does not support the PDBx/mmCIF format [30]. For users seeking a more user-friendly option, AMIGOS III offers a PyMOL plugin that calculates pseudotorsion angles and presents them in Ramachandran-like plots [17].

In this paper, we introduce RNAtango (RNA structure analysis in Torsion ANGle dOmain), a powerful web-based platform for comprehensive analysis of 3D structures of nucleic acid (RNA, DNA or their hybrids) molecules using trigonometric representation. RNAtango supports regular and modified nucleotides on input such as methylated nucleobases. The system processes structures in PDB or PDBx/mmCIF formats [31], accommodating Protein Data Bank entries [32] and user-uploaded files generated by RNA structure modeling tools. The web server calculates the angular parameters of an input structure from full-atom coordinates and performs a statistical analysis of these parameters. It excels in comparative analysis, applying torsion and pseudo-torsion angles to evaluate multiple structures simultaneously. Two primary comparison modes are available: cluster analysis to assess similarities among various models and reference-based comparison to measure deviations from a designated target. Advanced comparison and alignment techniques implemented in RNAtango include MCQ (Mean of Circular Quantities) [33] and LCS-TA (Longest Continuous Segment in Torsion Angle space) [34], both rigorously validated in the RNA-Puzzles contest for blind prediction of RNA 3D structures [35–39]. RNAtango, freely accessible at http://rnatango.cs.put.poznan.pl/, combines a user-friendly design with sophisticated analytical capabilities. Although there are few computational tools to determine selected angular parameters in RNA structures [40, 41], none offers as comprehensive processing and analysis of these parameters as RNAtango.

**Table 1. Torsion (TA) and pseudo-torsion (PA) angles defined for RNA.**

| TA | Atoms involved | TA | Atoms involved | PA | Atoms involved |
|---|---|---|---|---|---|
| $\alpha$ | O3'$_{n-1}$-P-O5'-C5' | $\chi$ | O4'-C1'-N1-C2 (pyrymidines) | $\eta$ | C4'$_{n-1}$-P-C4'-P$_{n+1}$ |
| $\beta$ | P-O5'-C5'-C4' | | O4'-C1'-N9-C4 (purines) | $\theta$ | P-C4'-P$_{n+1}$-C4'$_{n+1}$ |
| $\gamma$ | O5'-C5'-C4'-C3' | $\nu_0$ | C4'-O4'-C1'-C2' | $\eta'$ | C1'$_{n-1}$-P-C1'-P$_{n+1}$ |
| $\delta$ | C5'-C4'-C3'-O3' | $\nu_1$ | O4'-C1'-C2'-C3' | $\theta'$ | P-C1'-P$_{n+1}$-C1'$_{n+1}$ |
| $\epsilon$ | C4'-C3'-O3'-O | $\nu_2$ | C1'-C2'-C3'-C4' | | |
| $\zeta$ | C3'-O3'-P-O5'$_{n+1}$ | $\nu_3$ | C2'-C3'-C4'-O4' | | |
| | | $\nu_4$ | C3'-C4'-O4'-C1' | | |

## Design and implementation

### Torsion and pseudo-torsion angles in RNA structure

The three-dimensional structure of the RNA molecule is commonly stored in either PDB or PDBx/mmCIF format, which includes extensive metadata, experimental conditions, and atomic coordinates. Bioinformatics tools frequently use this spatial information to annotate RNA interactions, align structures, or assess their properties. These tasks often require calculating additional parameters or adopting a different representation, such as a trigonometric model that uses torsion angles to describe rotations around atomic bonds. RNA folds are characterized by several angular parameters. Six angles, $\alpha$, $\beta$, $\gamma$, $\delta$, $\epsilon$, and $\zeta$, describe a nucleotide backbone; the angle $\chi$ determines the glycosidic bond; five torsion angles, $\nu_0$–$\nu_4$, describe the ribose component (Table 1).

The ribose ring adopts two primary conformations: the envelope form, where four atoms are coplanar and one is out of a plane, or the twist form, featuring three coplanar atoms with two adjacent atoms displaced on opposite sides of the plane. These conformations can be distinguished based on a single parameter, the pseudorotation phase angle $P$ (Eq 1), which elegantly substitutes for the five highly correlated angles $\nu_0$–$\nu_4$. $P$, ranging from 0˚ to 360˚, precisely represents the envelope or twist modes, offering a concise yet comprehensive description of the ribose geometry. The usage of constants $\sin 36°$ and $\sin 72°$ in Eq 1 allows to define the C2'-exo-C3'-endo twist form at $P = 0°$. Every 18˚ increment in P shifts the ribose ring to a new conformation and two conformations differing by 180˚ in P have opposite signs for all their torsion angles, such as $P = 90°$ is O4'-endo and $P = 270°$ is O4'-exo.

$$P = \arctan \frac{(\nu_4 + \nu_1) - (\nu_3 + \nu_0)}{2\nu_2(\sin 36° + \sin 72°)} \tag{1}$$

RNA structure analysis frequently utilizes pseudo-torsion angles, defined between non-bonded atoms. Angles $\eta$, $\theta$, $\eta'$, and $\theta'$ (Table 1, Fig 1) have been effectively used in research to describe RNA folding. They provide a coarse-grained picture of the shape of the RNA backbone, allowing structural analysis similar to Ramachandran plots in protein bioinformatics [42].

### Angular metrics to score structure similarity

Torsion and pseudo-torsion angles can be used to assess structural similarities when comparing 3D structures. This approach involves calculating the differences between the corresponding angles in two structures, taking into account the periodic nature of angular values. Formula 2 satisfies the latter condition, ensuring that the minimum angular difference is

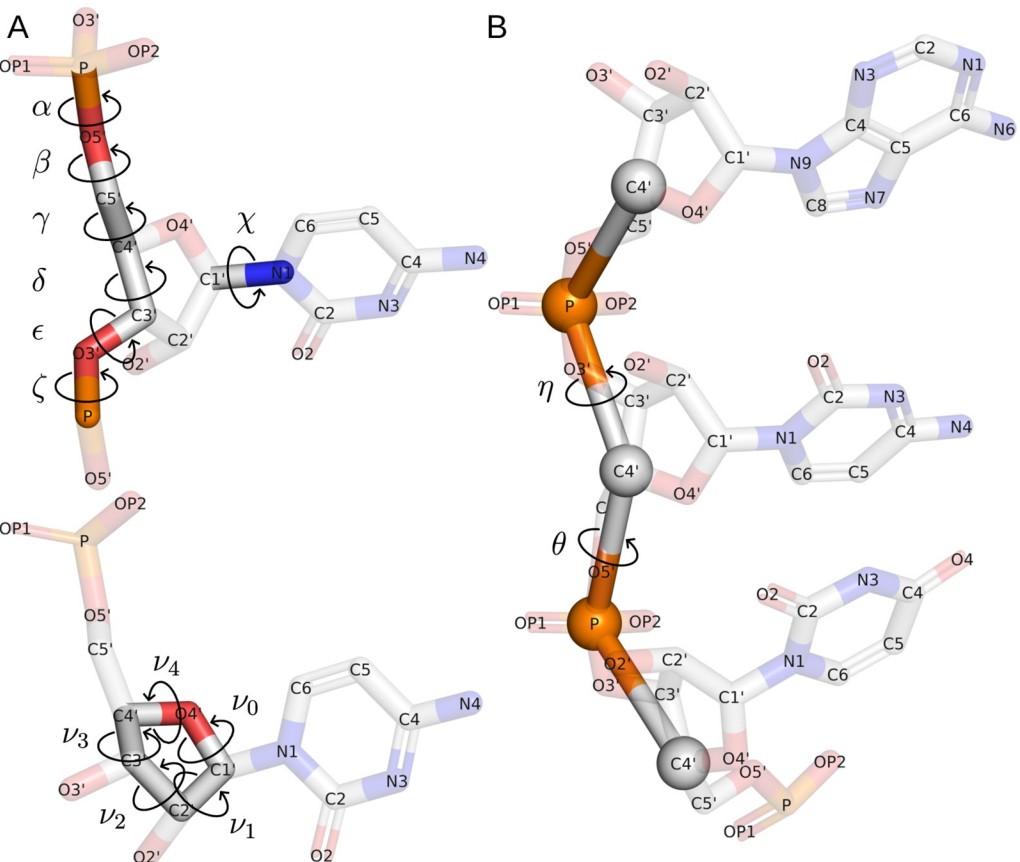

**Fig 1. Torsion angles (A) and pseudo-torsion angles (B) in RNA structure.**

always considered, regardless of direction.

$$\Delta(\angle_A, \angle_B) = \min(|\angle_A - \angle_B|, 360° - |\angle_A - \angle_B|) \tag{2}$$

After calculating individual angle differences, it is essential to aggregate them into a single meaningful dissimilarity metric. This is where the Mean of Circular Quantities (MCQ) comes into play [33]. MCQ is an elegant solution that maintains the data's circular nature while providing a single, easily interpretable value. To compute it, the three-step procedure (Fig 2) should be applied: (i) decompose each angular difference into vertical (sine) and horizontal (cosine) components; (ii) sum these components across all the differences; (iii) use the arctangent function to combine the sums into a final average difference. The MCQ over a set $D$ of angular differences can be expressed as:

$$MCQ(D) = \arctan \frac{\sum_{d \in D} \sin d}{\sum_{d \in D} \cos d} \tag{3}$$

This adaptable method allows for customization based on specific research needs. Typically applied to all torsion and pseudo-torsion angles to generate a comprehensive structural dissimilarity score, the MCQ can also be calculated for individual residues or selected subsets of angles and residues. This flexibility makes it invaluable for structural biologists and

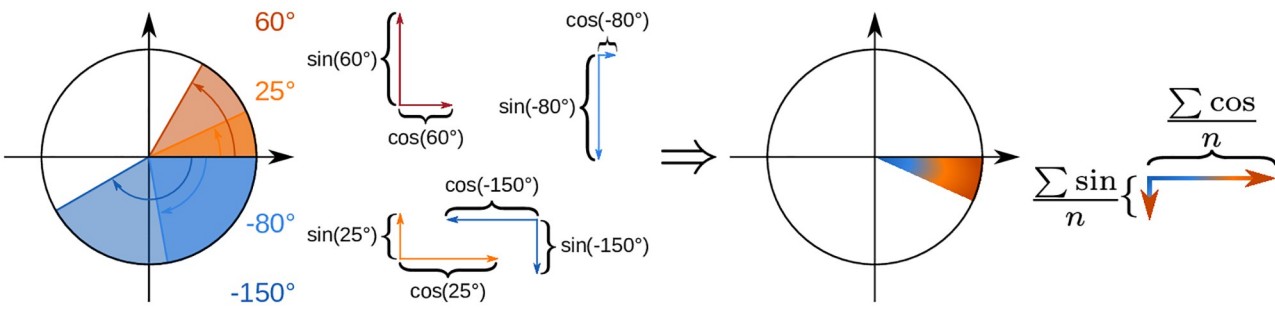

**Fig 2. The procedure for MCQ computation.**

computational chemists aiming to quantify and compare 3D molecular structures with precision and nuance. MCQ metric is also effective for clustering or aligning structural data. In LCS-TA (Longest Common Segment—Torsion Angles), we leverage this metric to identify the longest continuous segment between two structures that maintains an MCQ score below a specified threshold [34]. LCS-TA employs a sliding window in a divide-and-conquer algorithm to find the maximum length segment where the MCQ remains below a given value. The algorithm outputs the position and length of this segment, with the latter serving as a similarity measure. The results can be visualized to highlight the structural context of the matched segment.

Comparison of structures in the torsion angle space offers significant advantages, primarily because it is independent of superposition. Torsion angles remain constant regardless of translation or rotation, simplifying comparisons and facilitating machine learning model training. Furthermore, MCQ and LCS-TA can be customized based on angle types and analysis scope, making them versatile for both global and local dissimilarity measures. As global measures, they provide a single dissimilarity value; as local measures, they evaluate smaller fragments. Lastly, torsion angle-driven measures complement Cartesian-based ones, offering unique insights into structural similarity [43].

However, MCQ also has some disadvantages. As an averaging-based measure, it may inadvertently mask specific issues by balancing errors with correctly modeled regions. Currently, MCQ does not differentiate between helical regions and single strands, treating them equivalently. Helical regions, which have less flexibility in torsion angle distributions and are often accurately modeled, can result in lower MCQ values for helix-rich structures, even if the overall fold is incorrect. We acknowledge this limitation and are actively working on solutions. This complex task involves integrating 2D structural alignment with a weighted MCQ metric to account for models that fold differently from their reference structures.

Finally, it is important to note that comparative analysis based solely on angular parameters is not a substitute for evaluation using atomic coordinates, as the measures derived from these two data types do not correlate [14]. Therefore, to gain a more comprehensive understanding of the similarities and differences between the structures, it is recommended to use both methods.

## RNAtango web application

The RNAtango backend is a Java 17 application built using the Spring Boot framework. It interfaces with a PostgreSQL database and an Apache RabbitMQ message broker. The system employs WebSocket technology for interactive dual-channel communication with the

frontend. The frontend, developed with the React framework, is hosted on Nginx. All components are containerized, with RNAtango operating as a Docker Compose application. The system runs on a machine with 4 CPUs and 8 GB of RAM under the Ubuntu GNU/Linux operating system.

RNAtango supports three analytical scenarios (Fig 3). The *Single model* scenario allows users to input a single RNA structure, which can be (i) specified using a PDB ID, (ii) a locally uploaded file, or (iii) one of the provided examples. The users select a model, one or more chains, and a range of nucleotides for analysis, and submit the task. It is processed with real-time progress updates displayed on the RNAtango website. The results are presented in an interactive table containing all torsion and pseudo-torsion angles computed for the input

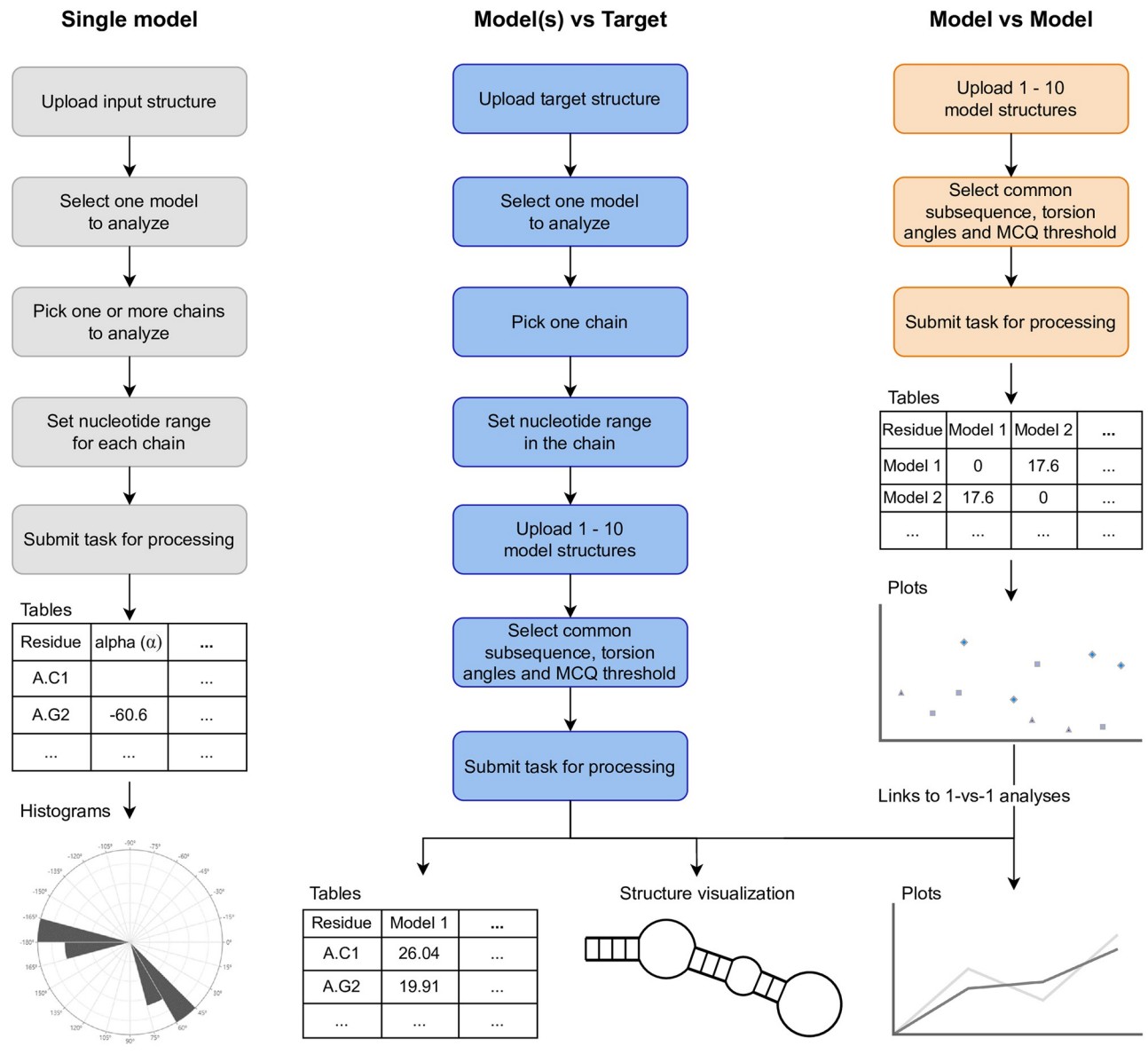

**Fig 3. Flowchart for the three scenarios supported in RNAtango.**

structure. The result page also includes angular histograms and statistics on the *syn* and *anti* conformations of the glycosidic bond.

The second scenario, *Model(s) vs Target*, focuses on a structural comparison using (pseudo) torsion angles. Users specify the target structure, select one model, one chain, and a nucleotide range, and then upload other models sequentially if needed. Up to 10 models can be handled in one task. RNAtango displays the longest common subsequence found for the selected nucleotide range in the target and in the sequences corresponding to the models. Users can configure which angles to use for comparison with the MCQ measure and set thresholds for the LCS-TA algorithm. The processing progress is clearly shown to the user. The result page features a table with per-residue MCQ values for each model, visualized on an interactive heatmap with the secondary structure context. The heatmap uses a color spectrum to depict differences: yellow for 0˚-15˚, light orange for 15˚-30˚, dark orange for 30˚-60˚, and red for >60˚. Additionally, the result page includes tables showing the percentage of residues with differences under specified thresholds and detailed results for each model, including angular differences, line plots of these differences along the chain, and VARNA-based [44] secondary structure visualization color-coded according to angular differences. A table with LCS-TA results is also provided, indicating the length and location of the longest continuous segment with the required maximum level of dissimilarity. Each entry in this table can be visualized in 3D using the embedded Mol* viewer [45].

The *Model vs Model* scenario is intended for cases where the target structure is unknown but the user seeks information about angular differences between several models. Users upload files with RNA structures and configure the settings for torsion angles and the LCS-TA threshold. Progress tracking is available during processing. The result page includes a dissimilarity matrix computed for all models, which is also used for data clustering. Its results are displayed on a dedicated dendrogram and graph diagram. Each pairwise model comparison links to a *Model(s) vs Target* view, where one model serves as the target, facilitating a deeper understanding of the differences between specific pairs of models.

In every scenario, the generated outputs are available for download in standard formats: CSV for tables and SVG for histograms. Each task is assigned a unique URL, allowing users to easily return to cached results.

## Results

In this chapter, we provide examples of how the RNAtango system works in each of the three available usage scenarios.

### Single model scenario

In 2004, Popenda *et al.* elucidated the first left-handed RNA double helix $(CGCGCG)_2$ using NMR spectroscopy (PDB id: 1T4X) [46]. This structure is notable for its highly irregular backbone, attributed to the distinct GpC and CpG steps, and is classified as Z-RNA, in contrast to the more common A-RNA double helix (Fig 4). Unlike Z-DNA, Z-RNA features base pairs that are much closer to the helical axis. The unique characteristics of the Z-RNA double helix can be understood through its torsion angles, as evidenced by the results displayed in RNAtango (Fig 4B–4E). Specifically, the guanosines exhibit *syn* glycosidic bonds ($\chi$ between 30˚ and 120˚), $\alpha$ in *gauche+* (between 30˚ and 90˚), $\gamma$ in *trans* (between 150˚ and 210˚), and $\zeta$ in *gauche-* (between 270˚ and 330˚). Conversely, the cytosines display *anti* glycosidic bonds ($\chi$ between –240˚ and 30˚), $\alpha$ in *trans*, $\gamma$ in *gauche+*, and $\zeta$ in *gauche+*. Despite these variances, the Z-RNA helix maintains canonical Watson-Crick-Franklin C-G base pairs.

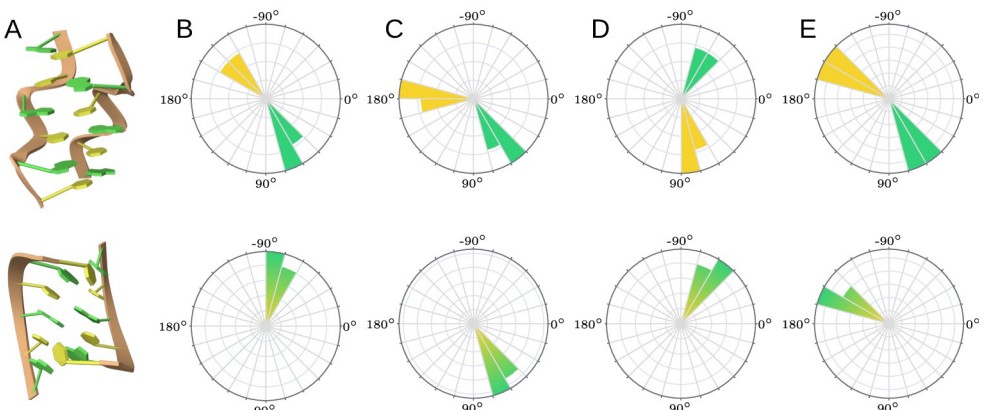

**Fig 4. Two (CGCGCG)$_2$ structures.** Top: a left-handed Z-RNA conformation (PDB id: 1T4X). Bottom: a right-handed A-RNA conformation (PDB id: 1PBM). (A) 3D visualization, (B-E) angular histograms for $\alpha$, $\gamma$, $\zeta$ and $\chi$ angles respectively.

## Model(s) vs Target scenario

Exonuclease-resistant RNAs (xrRNAs) are molecules predominantly found in viruses. Their unique structural features confer resistance to exonucleases, contributing to their pathogenicity. In 2016, Akiyama *et al.* resolved the three-dimensional structure of an xrRNA from the Zika virus [47]. This structure was targeted in RNA-Puzzles Round V (Puzzle 18) [48], with its prediction complexity attributed to a critical pseudoknot forming a knot-like motif, representing a higher-order entanglement of structural elements [49]. As an example, we selected seven models predicted in Puzzle 18. They were submitted by different modeling groups within the human prediction category, where expert knowledge and manual interventions are allowed to achieve optimal predictions. For the analysis, we chose model number 1 from each of the groups participating in this challenge. In the manuscript, we refer to these models as H1-H7, while the real group names are available in the S1 Table.

RNAtango facilitates the comparison of multiple models against a reference RNA structure by presenting results in a heatmap, where each residue in every model is color-coded based on an MCQ value computed per residue (see Fig 5 for heatmap and Fig 6A for the 2D structure

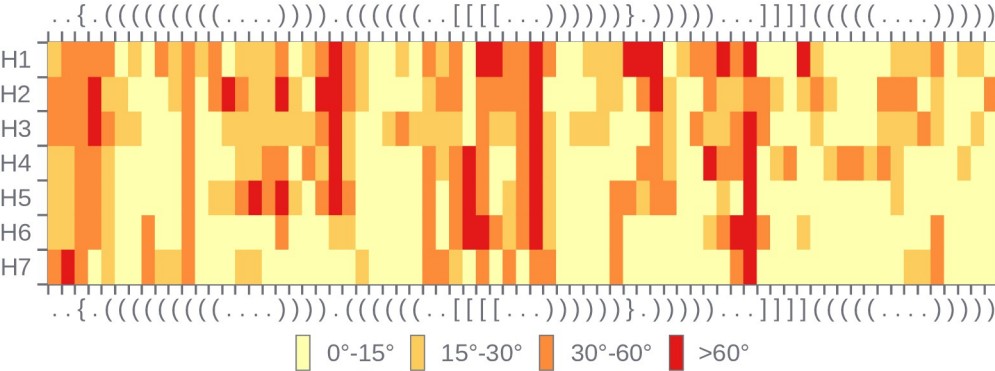

**Fig 5. Heatmap that shows per-residue MCQ values in *Model(s) vs Target* scenario of RNAtango, resulting from an analysis of selected predictions against the structure targeted in RNA-Puzzles 18.**

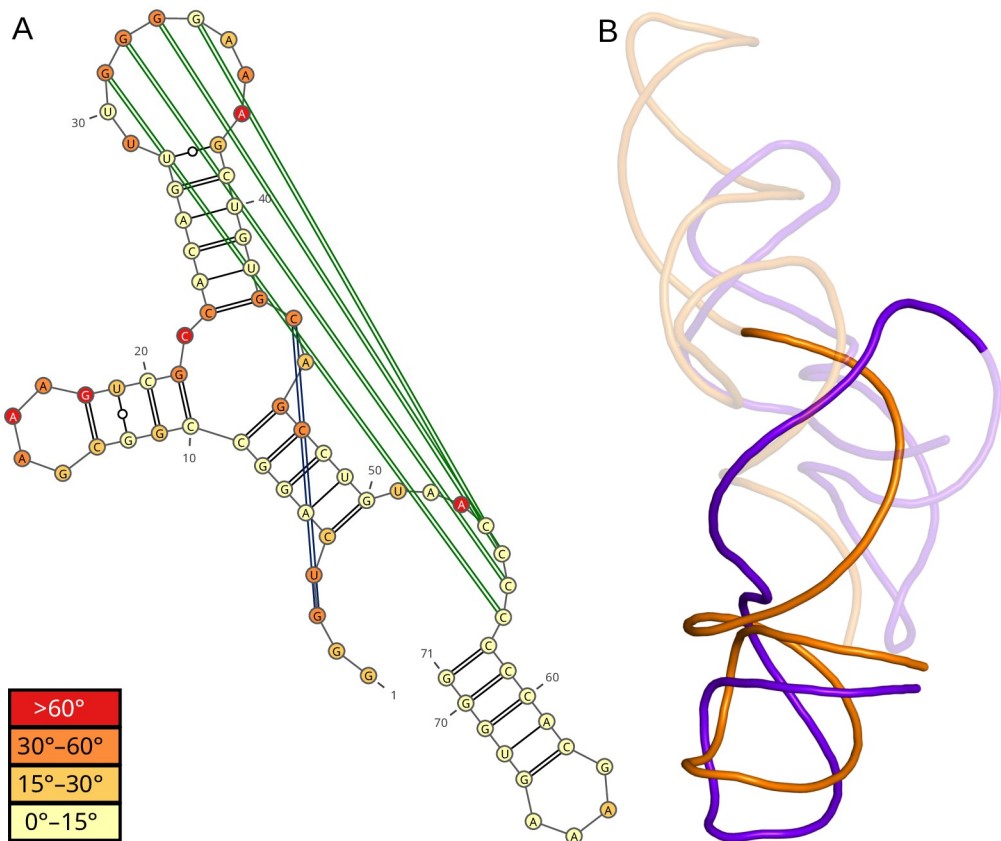

**Fig 6. Comparison of H5 model with the native structure in RNA-Puzzles 18.** (A) 2D structure color-coded according to per-residue MCQ values and (B) alignment of the longest continuous segments for which MCQ≤15˚— target in purple, model in orange.

visualization). The heatmap also includes information about the secondary structure of the target to provide additional context for model evaluation. In the case of the Zika virus xrRNA, its structure contains a second-order pseudoknot [50], which is a challenge for predictive algorithms. Therefore, the heatmap (Fig 5) reveals that single-stranded regions around the pseudoknots exhibit the greatest dissimilarities across models, with per-residue MCQ values frequently falling within the 30˚–60˚ or >60˚ ranges. In contrast, the hairpin at the 3'-end is the easiest to predict, often showing per-residue MCQ values in the 0˚–15˚ range. The rows of the heatmap highlight specific differences between models. For example, the first hairpin shows notable differences in the predictions submitted by H1-H5, but not in those by H6-H7 groups.

Beyond aggregate results, RNAtango provides detailed information about each model compared. Users can access a table of angular differences, a line graph, and a 2D structure visualization prepared with VARNA [44], all color-coded according to per-residue MCQ differences. This detailed view also identifies the longest continuous segment where the MCQ dissimilarity value remains below a specified threshold. For example, setting this threshold to 15˚ allows an assessment of the lengths of the segments in each model. Examining the model by the H5 group (Fig 6) highlights structural differences in this prediction. The 2D structure visualization shows that the most dissimilarities occur in single-stranded regions and pseudoknots, while stems generally resemble the target structure, except for the hairpin starting at nucleotide index 10. The LCS-TA algorithm identifies the longest continuous segment at the 3'-end. The

3D visualization confirms that the backbone trace in the model aligns closely with the target, despite a slight directional change before the last hairpin at the 3'-end.

### Model vs Model scenario

Many RNA modeling approaches involve an intelligent search within the conformational space, guided by a scoring function that assigns energy-like values to the 3D atomic arrangements. This guides the simulation toward low-energy states. Over the years, several scoring functions have been developed for RNA 3D structure modeling. Developers of these scoring functions typically utilize common benchmark data, which include selected native structures and various decoys—intermediate steps in the 3D simulation process of RNA. A notable benchmark collection was introduced by Capriotti *et al.* in 2011 [51].

To illustrate the capabilities of RNAtango, we selected a target and its nine decoys from the benchmark dataset. The chosen target is chain R from the U2 small nuclear RNA (PDB ID: 1A9N) [52]. We conducted the analysis in a *Model vs Model* scenario, comparing each input structure with the others. The resulting dissimilarity matrix (Fig 7) elucidates the structural relationships among the models. Notably, 1a9nR_M9 stands out as an outlier, showing

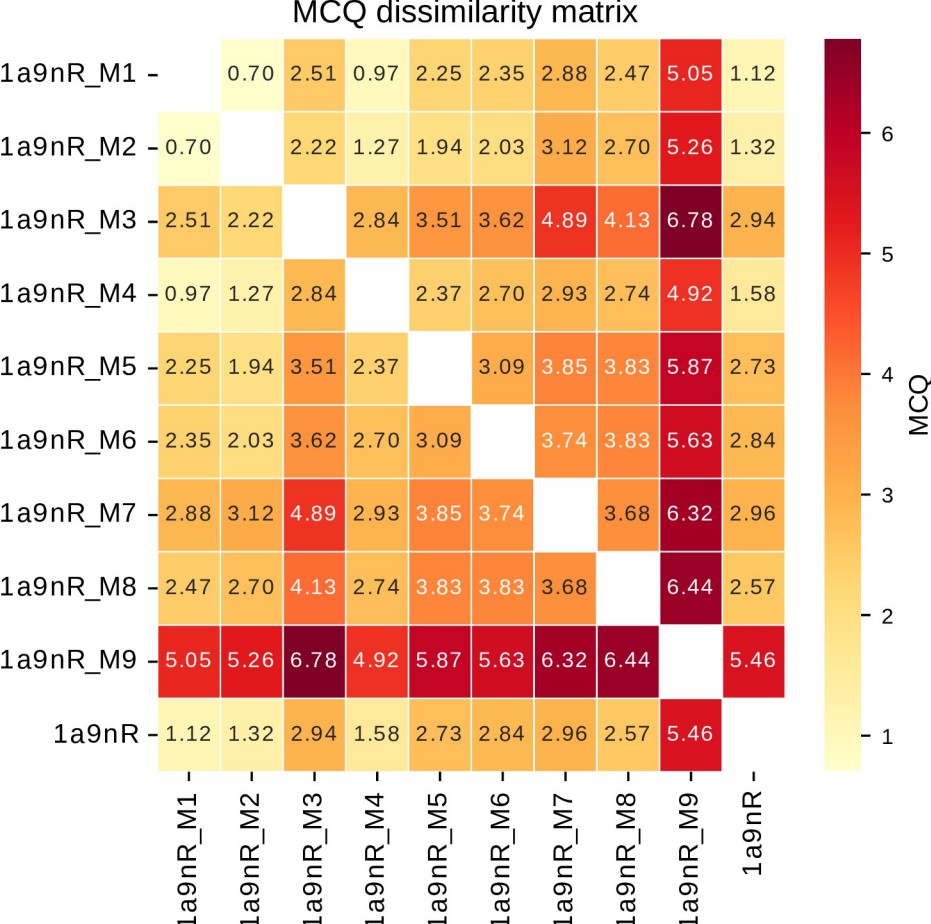

**Fig 7. The MCQ dissimilarity matrix illustrates the angular distances between nine decoy structures and the native configuration of U2 small nuclear RNA (PDB ID: 1A9N).**

significant dissimilarity from all other structures. 1a9nR_M1 model is most similar to the native structure, followed closely by 1a9nR_M2 and 1a9nR_M4.

The dissimilarity matrix serves as a robust input for various analytical methods. In RNAtango, we implemented a Classical Multidimensional Scaling (MDS) technique, also known as Torgerson or Torgerson-Gower scaling [53], transforming the dissimilarity matrix into a two-dimensional space. This transformation aims to preserve the relative distances between points as accurately as possible according to the original matrix data. It is important to note that our data is angular, which introduces some inevitable errors when using MDS to maintain Euclidean distances. Despite these limitations, the approach remains valuable for visually assessing the distribution of points. Our experiment confirmed this utility, as the 1a9nR_M9 structure was distinctly isolated from the other points, while the native structure clustered closely with models exhibiting low MCQ (Fig 8).

Clustering analysis provides an additional method to gain insight into the comparative data of RNA models. Among the various clustering algorithms that accept dissimilarity matrices as input, we chose to implement k-Medoids in RNAtango because of its advantages over the popular k-Means approach. [54]. Unlike k-Means, which require Euclidean data, k-Medoids operate directly on the dissimilarity matrix, selecting cluster centers from existing datapoints. We

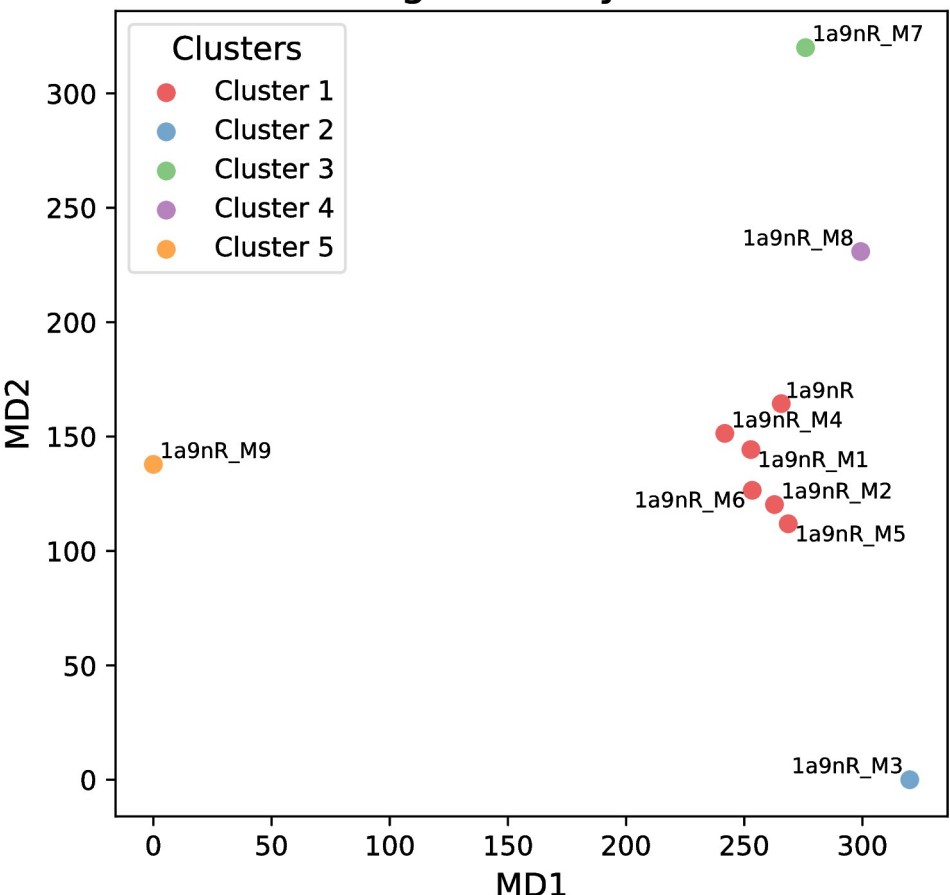

**Fig 8. Multidimensional scaling of the MCQ dissimilarity matrix with color-coded results of k-Medoids clustering for nine decoy structures and the native configuration of U2 small nuclear RNA (PDB ID: 1A9N).**

applied the k-Medoids algorithm with $k = 5$ to our dataset, and the results are illustrated in Fig 8. The clustering outcome reveals a notable pattern: one cluster encompasses the native structure along with five decoys exhibiting high similarity to it. Each of the remaining four clusters contains a single model. This clustering arrangement aligns well with the input dissimilarity matrix, as each isolated model in its own cluster demonstrates strong dissimilarity to all other models. Clustering offers a valuable perspective on the structural relationships among compared RNA models, effectively grouping similar structures and isolating distinct ones. Such an analysis can help identify representative structures and outliers within the dataset, potentially guiding further investigations into structural variations and their biological significance.

## Future directions

Future developments for RNAtango include adding functions to read and process (generate data statistics, perform comparison) RNA structures uploaded in trigonometric format, such as angular data predicted by SPOT-RNA-1D [21] or RNA-TorsionBERT [23]. However, this will require defining a standard format for such data and/or implementing a new module to transform possible formats into one, accepted by RNAtango. Additionally, we plan to extend the single-structure scenario by including an option to process molecular dynamics (MD) trajectories. This will enable the generation of histograms that illustrate changes in torsion angles throughout the simulation, providing deeper insights into the dynamics of RNA molecules.

## Supporting information

**S1 Table. Actual names of models used to exemplify *Model(s) vs Target* scenario.**
(XLSX)

## Author Contributions

**Conceptualization:** Marta Szachniuk, Tomasz Zok.

**Data curation:** Marta Szachniuk, Tomasz Zok.

**Formal analysis:** Marta Szachniuk, Tomasz Zok.

**Funding acquisition:** Tomasz Zok.

**Methodology:** Marta Szachniuk, Tomasz Zok.

**Software:** Marta Mackowiak, Bartosz Adamczyk.

**Supervision:** Marta Szachniuk, Tomasz Zok.

**Validation:** Marta Mackowiak, Bartosz Adamczyk, Marta Szachniuk, Tomasz Zok.

**Visualization:** Marta Mackowiak, Bartosz Adamczyk, Tomasz Zok.

**Writing – original draft:** Marta Szachniuk, Tomasz Zok.

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
