## [Decision Letter · Decision Letter 0]

30 Aug 2024

Dear Marta,

Thank you very much for submitting your manuscript "RNAtango: analysing and comparing RNA 3D structures via torsional angles" for consideration at PLOS Computational Biology. As with all papers reviewed by the journal, your manuscript was reviewed by members of the editorial board and by several independent reviewers. The reviewers appreciated the attention to an important topic. Based on the reviews, we are likely to accept this manuscript for publication, providing that you modify the manuscript according to the review recommendations.

Sincerely,

Shi-Jie Chen

Academic Editor

PLOS Computational Biology

Arne Elofsson

Section Editor

PLOS Computational Biology

Reviewer's Responses to Questions

**Comments to the Authors:**

Reviewer #1: In this paper, the author developed a web server for RNA structure analysis according to backbone torsional angles. The server provides three scenarios of analysis, single model, model vs target and model vs model, which are very useful for RNA structure analysis, especially for the assessment of RNA structure prediction. The manuscript was written clearly, but I have a few questions about the work.

1, In equation 1, the author defined angle P to describe the structure status of the ribose ring. Is there any geometric significance of this angle? What is the meaning of sin 36˚ and sin 72˚? Why not just use ν2 to represent the status of ribose ring, I think it’s more simple and intuitive.

2, In the last paragraph of page 4, the author mentioned the benefits of comparison of RNA structure in torsion angle space, what about the disadvantages? How reliable it is comparing with RMSD based method?

3, The website is not stable. Sometimes I got the error message “Something went wrong, please try again” after submitting, and sometimes I just can’t open the website. After two days of trying, I finally succeeded once, and then the next day it didn’t work again.

Reviewer #2: The manuscript by Mackowiak et al. describes RNAtango, a web server that offers an analysis of the torsion angles of RNA molecules. The methodology is described in detail and the three examples of intended use are helpful in understanding just exactly what information users of RNAtango can obtain. The manuscript is clearly written, but would benefit from a few clarifications and a more detailed discussion of certain points.

Specific comments:

1. Since (and as mentioned in the manuscript) there are a few computational tools that already determine angular parameters of RNA structure, I would like to see an expanded discussion on how RNA tango compares to those tools, what is the main difference and what additional value it brings. Moreover, other tools in general take as input DNA structures or mixtures of NA molecules as well. Still, I am missing the information if RNAtango is capable of processing other nucleic acids. Can it handle RNA molecules with chemical modifications?

2. As one of the reasons for the use of torsion angle analysis, as specified in the introduction (line 24+), the authors suggest that the RNA folding process depends on the flexibility of the structure determined by angular parameters. I find this description misleading or perhaps just unfortunately phrased since it is the other way around and the torsion angles are restrained and flexibility is limited by energetic conformational minima stabilized with hydrogen bonds and stacking interactions.

3. In a single model mode, is it possible to display regular regions for specific RNA form alongside the analyzed ranges in angular histograms? I believe this would be helpful for the users, to see how much certain torsion angles deviate from the typical range. Related to this, please provide such region for A-form RNA in Figure 4.

4. For Figure 5 please add a schematic secondary structure of Zika virus xrRNA next to the heatmap to more easily relate dot-bracket notation of the heatmap with topology. What is the reason for not presenting H1-H7 in numerical order in the heatmap?

5. Please avoid using abbreviations (MCQ and LCS-TA) in the abstract.

Reviewer #3: RNAtango is a work that enables a complete and comprehensive description of RNA torsional angles from a webserver. With this work, the authors propose the study of RNA structures in three scenarios: single mode, model(s) vs target and model vs model. They strengthen the study with the add of associated metrics for structural quality assessment, the secondary structure as well as clustering with dissimilarity matrix.

The work is validated with examples for each scenarios, showing the promising interests of such tool.

The interface is clean and user-friendly, while also being well documented.

Minor concerns:

- Fig 7: the size of the numbers might be too high and make the figure less readable.

- Would it be available and interesting to add for the histogram an option to compare the histogram with the available data from the PDB?

- Is it possible for users to apply the model vs model with more than 10 models ? What if there are 1000 structures to rank?

- It could be interesting to add the correlation with RMSD or TM-score (distance-based metrics) with MCQ to highlight the correlation between these distance-based metrics and the torsional-based approach.

- Is it a way to download the angles predicted with single mode for a user?

**Have the authors made all data and (if applicable) computational code underlying the findings in their manuscript fully available?**

Reviewer #1: Yes

Reviewer #2: Yes

Reviewer #3: **No: **The authors provide a web server, nor a source code. About the data, they seem to be not provided (we didn't success to find them).

PLOS authors have the option to publish the peer review history of their article (what does this mean?). If published, this will include your full peer review and any attached files.

Reviewer #1: No

Reviewer #2: No

Reviewer #3: No

Figure Files:

Data Requirements:

Reproducibility:

References:

---

## [Editor Report · Decision Letter 1]

18 Sep 2024

Dear Marta,

We are pleased to inform you that your manuscript 'RNAtango: analysing and comparing RNA 3D structures via torsional angles' has been provisionally accepted for publication in PLOS Computational Biology.

Best regards,

Shi-Jie Chen

Academic Editor

PLOS Computational Biology

Arne Elofsson

Section Editor

PLOS Computational Biology

---

## [Editor Report · Acceptance letter]

27 Sep 2024

PCOMPBIOL-D-24-01228R1 

RNAtango: analysing and comparing RNA 3D structures via torsional angles

Dear Dr Szachniuk,

I am pleased to inform you that your manuscript has been formally accepted for publication in PLOS Computational Biology. Your manuscript is now with our production department and you will be notified of the publication date in due course.

With kind regards,

Olena Szabo
